# Survey of Text Mining Techniques Applied to Judicial Decisions Prediction

Olga Alejandra Alcántara Francia [1,*] , Miguel Nunez-del-Prado [2,3] and Hugo Alatrista-Salas [4,5,*]

1   Faculty of Law, Universidad de Lima, Lima 15023, Peru
2   Peru Research, Development and Innovation Center (Peru IDI), Lima 15076, Peru
3   Instituto de investigación de la Universidad de Andina del Cusco, Cusco 080104, Peru
4   Escuela de Posgrado Newman, Tacna 23001, Peru
5   Facultad de Ciencias e Ingeniería, Pontificia Universidad Católica del Perú, Lima 15088, Peru
*   Correspondence: oalcanta@ulima.edu.pe (O.A.A.F.); halatrista@pucp.edu.pe (H.A.-S.)

**Abstract:** This paper reviews the most recent literature on experiments with different Machine Learning, Deep Learning and Natural Language Processing techniques applied to predict judicial and administrative decisions. Among the most outstanding findings, we have that the most used data mining techniques are Support Vector Machine (SVM), K Nearest Neighbours (K-NN) and Random Forest (RF), and in terms of the most used deep learning techniques, we found Long-Term Memory (LSTM) and transformers such as BERT. An important finding in the papers reviewed was that the use of machine learning techniques has prevailed over those of deep learning. Regarding the place of origin of the research carried out, we found that 64% of the works belong to studies carried out in English-speaking countries, 8% in Portuguese and 28% in other languages (such as German, Chinese, Turkish, Spanish, etc.). Very few works of this type have been carried out in Spanish-speaking countries. The classification criteria of the works have been based, on the one hand, on the identification of the classifiers used to predict situations (or events with legal interference) or judicial decisions and, on the other hand, on the application of classifiers to the phenomena regulated by the different branches of law: criminal, constitutional, human rights, administrative, intellectual property, family law, tax law and others. The corpus size analyzed in the reviewed works reached 100,000 documents in 2020. Finally, another important finding lies in the accuracy of these predictive techniques, reaching predictions of over 60% in different branches of law.

**Keywords:** judicial prediction; legal tech; legal prediction; machine learning; natural language processing; deep learning

## 1. Introduction

Recently, we have witnessed the application of Artificial Intelligence (AI) in different domains, such as health, computer vision, privacy and law. This trend was due to the increasing data gathered worldwide. This amount of rich data allows the use of AI-based techniques. Particularly, we have seen different efforts to collect and make available legal corpora in the law domain, such as terms-of-service agreements [1], privacy policies (Privacy policies https://usableprivacy.org/data, accessed on 13 March 2022), English contracts manually annotated [2], legislative documents from The EU's public document database [3], sentences from German rental agreements [4], court decisions on cybercrime and trade secrets [5], as well as Australian legal cases from the Federal Court of Australia (FCA https://archive.ics.uci.edu/ml/datasets/Legal+Case+Reports#, accessed on 13 March 2022), Danish Legal monolingual corpus (Danish Legal corpus elrc-share.eu/repository/browse/danish-legal-monolingual-corpus-from-the-contents-of-the-retsinformationdk-web-site, accessed on 20 March 2022), Japanese-English Legal Parallel Corpus (Japanese-English Legal Parallel Corpus https://datarepository.wolframcloud.com/resources/Japanese-English-Legal-Parallel-Corpus, accessed on 15

March 2022), and British and Irish Legal documents (British and Irish Legal Information Institute https://www.bailii.org/, accessed on 20 March 2022) and Human rights [6]. These sources of information enable the use of text mining to identify facts, relationships and assertions hidden in the vast amounts of textual data and documents. Text mining uses various methods to process text, the most important being Natural Language Processing (NLP). NLP, also known as computational linguistics, is a subfield of AI that learns, understands, recognizes, and produces human language content [7]. NLP techniques have been used in machine translation, word sense disambiguation, summarization, syntactic annotation and named entity recognition [8]. Specifically, NLP techniques have been used in different legal fields, demonstrating that AI can provide valuable support in legal work [9–11]. We mainly focused on Judgebot, which is the application of computational linguistic techniques for rule decision prediction in different legal domains, contexts, and languages.

In the industry, applying AI to certain processes is more accurate, faster, and cheaper than using humans. Thus, if we transpose these benefits to the legal system, using AI could reduce the time and costs of justice administration. Accordingly, technology is already transforming the justice system. For instance, in British Columbia, Canada, a new Civil Dispute Tribunal is intended to operate using an online platform to allow disputants to make initial contact and starts proceedings online. Furthermore, the Northern Ireland Courts and Tribunal Service now provides an online process for small claims. The UK Civil Justice Council ('CJC') report on 'Online Dispute Resolution for Low-Value Civil Claims' recommends the establishment of a new dedicated internet-based court service for civil disputes worth less than GBP 25,000. Furthermore, the AI Solicitor collects data and analyzes relevant case law (ROSS intelligence). Another example is the Estonian Ministry of Justice, which created a robot judge capable of resolving disputes of less than EUR 7000. Consequently, AI-based judgment is more likely to be used in small cases [12,13]. Therefore, the advantages of implementing such technologies include reduced case complexity and assistance with advice and suggestions [14]; courts can use AI to multiply the workforce to achieve more work with limited resources [15]. Thus, Judgebot reduces individual human biases and guarantees the impartiality of the process [16]. Consequently, implementing AI in the legal domain will be the next breakthrough in the legal domain. Furthermore, robot judges will change how humans and society make decisions [17]. Nevertheless, interpretability and fairness still exist. The former is concerned with the explanation of AI black-box models in the process of judgment. The latter is concerned with model learning over biased data, resulting in discriminatory models.

AI is used to improve public services decision-making, particularly, to manage many citizen contacts, identify financial reporting fraud, and enable governmental responses in assessing cyber-attacks [18]. However, some challenges regarding bias in AI due to incomplete and inaccurate data reflecting historical structural inequalities exist. Furthermore, the opaqueness created by the Machine Learning (ML) algorithm's internal rules determining the input to output is a challenge. Finally, when an AI decision is made (the machine, the creators of the machine, the coders, the managers who decided to deploy the machine), responsibility or liability must be attributed.

ML techniques have also been used to combat crime. The survey of Henrique et al. [19] summarized the proportion of used learning techniques, such as unsupervised learning (38.53%), supervised learning (30.28%) and association rules (17.43%). In addition, the authors reported the most used algorithms such as K-Means (14.39%), K-Nearest Neighbors (KNN) (11.36%) and Apriori (9.85%).

Based on the legal technology domain, Park et al. [20] present a survey focused on legal research and predictive analytics, as well as litigation data mining. The authors showed the proportion of studies produced in different regions, such as Asia (50%), Europe (31%), North America (11%), South America (6%) and Africa (2%). Accordingly, these studies focus more on ruling prediction using supervised learning, such as Artificial Neural Networks (ANNs) and Deep Learning (DL). In addition, the authors highlight all the challenges in structuring raw legal text for consumption by a supervised learning algorithm using NLP

techniques. In the same vein, Rosili et al. [21] survey gathered studies on court decision prediction in the literature. They used the publication standard Reporting Standards for Systematic Evidence Syntheses (ROSES) to collect different studies. The authors conclude that the Support Vector Machine (SVM) is the most used algorithm for this problem, with an accuracy of 80–91%. Sukanya and Priyadarshini [22] recently analyzed attention models on the ruling decision prediction. Specifically, they focus on encoding and decoding architecture using the attention mechanism of the transformer model. In a previously mentioned paper, the researchers have reviewed different previous studies on different DL models applied to the prediction of sentences in Human Rights from the European Court of Human Rights (ECtHR). They also analyzed documents from the different Chinese online judicial processes. Accordingly, the authors demonstrated that AI helps reduce procedural deadlines by 70%. Thus, integrating hierarchical attention neural network models with the adjusted transformer concept provides efficient quality- and time-based improvement in judgment prediction. However, further investigation is required to enhance multi-label classification for complex case facts with multiple defendants and charges. Future studies that focus on summarizing legal judgments, curating legal data, and simplifying legal documents will be very useful to the legal community.

Despite the significant advances in ML and DL techniques, there are still some challenges to overcome. For instance, to increase the predicting court's ruling decision accuracy [21] to improve Judicial systems. In addition to prediction, there are some concerns about the explainability of how these models make decisions and the potentially biased data used to train such models [23,24]. Consequently, using ML techniques to predict decision rules requires explanations about the features considered in making the decision. Furthermore, litigants have the right to receive explanations of the decision made. Thus, Atkinson et al. [25] describe the different techniques to explain the law, such as the explanation by example, explanation using rules, and hybrid systems. The authors emphasize the excellent performance and the need for new techniques to explain DL based on techniques for ruling prediction. Regarding fairness, Pessach and Shmueli [26] present a literature review to measure and avoid biases in the dataset for learning, such as biases due to missing data, biases from algorithm objectives and biases due to the use of proxy attributes instead of sensitive private data (e.g., gender, race, etc.).

This study reviews the most recent literature on ML- and DL-based techniques for rule decision prediction. In addition, the text representation techniques used for transforming legal text into a suitable form input to the learning algorithms are examined. Thus, we divide the reported studies based on the different types of law, such as Criminal Law, Administrative Law, Constitutional Law, Human Rights and Private Law, as shown in Figure 1, which depicts the classification of the branches of public and private law. In the reviewed studies, the viability of using these technological tools to predict the occurrence of illicit acts, administrative infractions, or Human Rights violations was demonstrated.

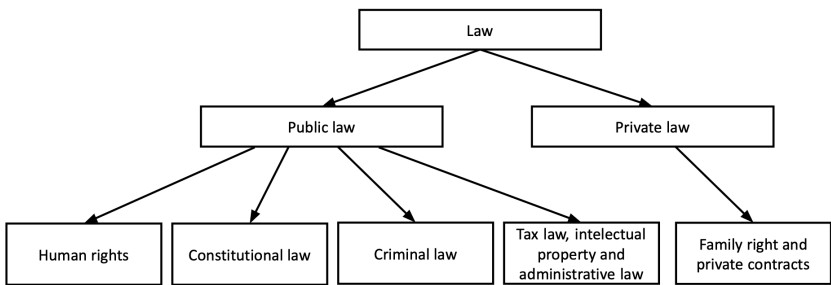

**Figure 1.** Survey classification by types of law.

The remainder of this study is organized as follows. First, Section 2 describes the methodology for collecting scientific studies. Section 3 demonstrates related studies focusing on five main topics. Then, Section 4 describes the analysis of the techniques, representation models, and metrics for judicial decision predictions. Section 5 discusses the

main findings of the literature review. Finally, Section 6 concludes the study and discusses future research directions.

## 2. Materials and Methods

This section details the methodology used to gather different scientific studies for the literature review. Accordingly, the building of a literature review is a response to a particular methodology. This step searches for all resources containing formal, informal, written, oral, or multimedia information on the problem under consideration. There are several methodologies for creating a literature review, but the most commonly used in academia are systematic and narrative reviews. The first methodology used in this study was inspired by [9,10]. The Systematic Literature Review SRL [27] begins by selecting a topic to study and transforming the project's objective into questions that the literature review will attempt to answer. The questions we aim to answer are summarized in Table 1.

**Table 1.** Research questions.

| Num. | Question |
| --- | --- |
| Q1 | Which law type applies judicial decision prediction through ML or DL? |
| Q2 | Which NLP and ML techniques were used to examine textual data associated with the legal text? |
| Q3 | What are the most used techniques in ML and DL to predict judicial decisions? |
| Q4 | Which countries apply or implement judicial decision prediction through ML or DL? |
| Q5 | Which quality measures are used to validate decision prediction? |
| Q6 | Which law type uses the most judicial decision prediction through ML or DL? |

Later, the inclusion and exclusion criteria for the studies to be reviewed are specified. In this study, we used keywords and search strings as inclusion criteria in the literature searches. The following list contains these words.

- Judicial decision prediction
- Ruling decision prediction
- Court decision prediction
- Robot judge
- Judgebot for ruling decision prediction
- Natural language processing + Human Rights
- Natural language processing + constitutional law
- Natural language processing + ruling prediction of Supreme Courts
- Natural language processing + Tax Law
- Natural language processing + intellectual property
- Natural language processing + Administrative Law
- Natural language processing + Criminal Law
- Deep learning + Judicial decisions prediction
- NLP + Human Rights
- Private law + Machine Learning + Deep Learning

Inclusion criteria also included the search language that was limited to English. In addition, we added a temporal restriction to filter only articles published after 2000. Thus, we developed a search plan that would include all types of literature. We chose Google Scholar as our search engine, which is a meta-search engine that includes articles from other search engines, such as IEEExplore, Scopus, Sciencedirect, WoS and ACM Digital Library. Therefore, Google Scholar is the most comprehensive academic search engine [28]. This document prioritizes the primary and secondary literature instead of the grey one. The Google Scholar meta-search engine sorts documents based on relevance; thus, we

selected the first 20 documents from each search string as the strategy. This preselection resulted in 140 documents. Finally, after a quick reading of the abstract and conclusions of the preselected documents, 45 documents were chosen and grouped into five groups.

Similar to our proposal, some authors review the literature about employing AI techniques in the legal domain. However, the differences between them and our contribution make it a recommended reading for legal practitioners working on these issues. For instance, Ref. [21] proposes an SRL of studies concerning the prediction of court decisions via machine learning methods, but the study does not consider deep learning methods. Furthermore, Ref. [29] studies the statistical limitations concerning three main problems in the classification of law and economics issues: anti-classification, classification parity and calibration. Similarly, the authors of [30] describe 17 legal applications. In contrast to our study, this paper only concentrates on applications and frameworks to help law practitioners. Likewise, the authors of [10] analyze various deep learning methods applied to the legal domain. The authors grouped the documents into three categories: legal data search, legal text analytics and intelligent legal interfaces. This study only concentrates on deep learning methods, unlike our proposal, which includes machine learning methods. Furthermore, in [31], the authors review machine learning algorithms applied in the legal domain. Even though various algorithms were studied, the authors concentrate only on interpretable models, prediction interpretation and justification. Lastly, in [32], the authors use the systematic literature review to perform a biometric analysis of 69 articles related to public law. The main results show a growing number of contributions from 2015 and an interaction of AI topics such as explainable machine learning techniques and decision support systems, among others. In contrast to the contributions mentioned above, our study includes machine and deep learning techniques. Furthermore, it includes a quantitative analysis using the main characteristics of the reviewed documents. Finally, we group contributions into five categories suggested by legal practitioners.

## 3. State-of-the-Art

In this section, we describe different works modeling the legal text representations and implementing algorithms to predict legal decisions in different law domains, such as Human Rights, Constitutional Law and Decision Prediction of Supreme Courts, Tax Law, Intellectual Property, and Administrative Law and Criminal Law, dealing with corpora in different languages such as Chinese (C), English (E), Farsi (Fa), French (F), German (G), Iranian (I), Philippine (Ph), Portuguese (P), Spanish (S) and Turkish (T).

### 3.1. Human Rights

Several initiatives have been proposed to study the legal corpus constituted by the ECtHR judgments. The focus of the study is on data mining analysis techniques for predicting violations or infringements of the European Convention on Human Rights (ECHR) articles.

For instance, in 2016, a group of American and British law and computer science researchers developed a predictive algorithm for the ECtHR [33]. This experiment was the first to train the SVM model for predictive analysis of the textual content of court decisions. The algorithm predicted whether there had been a violation of Article 3 (prohibition of torture and inhuman and degrading treatment), Article 6 (right to a fair trial), and Article 8 (respect for family and private life) of the ECHR. The algorithm was developed to predict violations of these articles exclusively based on information extracted from the text of the judgments concerning the description of the facts, applicable laws and the parties' arguments. The procedure, facts, circumstances and the relevant law of the cases were extracted from 584 decisions. Furthermore, they use a law violation of a given article of the Convention. The proposal achieves between 62% and 79% of accuracy in the decisions of the cases studied. An important conclusion was that the most relevant information for achieving the results was the description of the facts included in the circumstances of the case variable.

In the same vein, Medvedeva et al. [34] used 11532 ECtHR judgment documents to train an SVM linear classifier to predict future judicial decisions. A prediction accuracy of 75% was obtained for the violation of nine ECHT articles. The approach emphasizes the potential of ML techniques in the legal field. However, they showed that predicting decisions of future cases based on past cases negatively impacts performance (average accuracy range of 58–68%). Nevertheless, they demonstrated that relatively high classification performance could be achieved (average accuracy of 65%) by predicting outcomes solely based on the surnames of the judges hearing the cases.

Collenett, Atkinson, and Bench–Capon performed another study on the same dataset. They used the ANGELIC methodology to predict the outcomes of cases [35]. The experimental results were successful because they could predict court decisions with high accuracy (an average of 79%). The study used a dataset comprising 10 cases related to Article 6 of the ECtHR. The study concludes that Prolog can predict decisions with high accuracy and that this model can be adapted as the law evolves.

Furthermore, Liu and Chen [36] used 584 documents to compare five ML techniques—k-NN, logistic regression, bagging, random forests and SVM—to predict ECtHR judicial decisions. The authors used spectral clustering and N-grams to extract textual representations of topics, and they concluded that the SVM outperforms the other models.

The authors of [37] also predict law violation and non-violation using auto-sklearn to build models for 12 articles in the ECHR. The authors used n-grams, word embeddings (echr2vec) and doc2vec for the vectorization task for training Gradient Boosting (GB), Random Forest (RF), Stochastic Gradient Descent (SGD), Decision tree (DT), and Quadratic Discriminant Analysis (QDA). The authors obtained an average of 68.83% accuracy for the different models.

In a similar vein, Medvedeva et al. [38] propose a system to automatically predict judicial decisions using nine articles of the ECHR. Specifically, they propose an approach to automatically predict two possible classes, such as 'violation' of an article and 'non-violation'. After solving the unbalanced-classes problem, they vectorize the documents using the Term frequency-inverse term frequency (TF-IDF) normalization. Then words or tokens were grouped into bigrams, 3-grams, and 4-grams to improve the classification task using the SVM technique. They used 356 documents for Article 3, 746 for Article 6 and 350 for Article 8 for the training step, while the testing set was split by two years. Thus, the test set for 2014–2015 is 72, 80 and 52, and for 2016–2017, the number of documents is 140, 90, and 56 for articles 3, 6, and 8, respectively. In this manner, a corpus of 1942 documents was built. The results showed that the best model correctly predicts decisions between 66% and 77% of the cases.

Furthermore, Ref. [39] compares four algorithms for classifying European Court of Human Rights documents, specifically Articles 3, 6 and 8. The algorithms compared were SVM, KNN, Ensembles and gradient boosting through the accuracy measure and the 10-fold cross-validation strategy. Documents were vectorized using Bag-of-Words (BoW) technique followed by spectral clustering. The main result shows that the Ensemble classifier performs best in law documents.

Finally, in [40], the authors compare various standard ML algorithms on corpus extracted from the ECtHR in this effort. The authors propose a complete ETL pipeline to generate the benchmark scenarios to meet this goal. This pipeline includes the Bag-of-Words and TF-IDF documents vectorization, as well as the use of n-grams before the binary classification task. Several ML algorithms were compared, including Decision Tree (DT), AdaBoost with DT, Bagging with DT, Naive Bayes (NB) (Bernoulli and Multinomial), Ensemble Extra Tree, Extra Tree, Gradient Boosting, K-Neighbors, SVM (Linear SVC, RBF SVC), Neural Network (Multi-layer Perceptron) and Random Forest. The algorithms' performance was compared using quality measures, such as the accuracy, F1-score, and Matthews correlation coefficient.

### 3.2. Constitutional Law and Ruling Prediction of Supreme Courts

For the constitutional law and ruling prediction of the Supreme Courts, this study relies on sentences issued by the Supreme Courts of various countries, such as Turkey, the Philippines, the United States, Spain and France. The sentences used in the experiments are related to debates over property rights, adoption, freedom and so on. The documents, in some cases, were taken from institutional websites, such as CENDOJ (for the case of adoption in Spanish courts) or a compilation of sentences from constitutional courts (Turkey) or Cassation (France).

For instance, Ruger et al. [41] developed an algorithm that can predict the individual votes of the nine justices as well as the final direction of the court's decisions, that is, confirmation or revocation in the U.S. Supreme Court. This experiment was conducted with information obtained from the court for two years (2002–2003), and 628 cases were analyzed. The algorithm was based on a classification tree of the following six variables: the federal circuit in which the case originated, thematic area, type of plaintiff, type of defendant, ideological direction, and whether or not the constitutionality of a rule or practice was challenged. The results obtained were compared with the predictions made by a group of academics and lawyers. The following results were obtained: of 78 cases reviewed in 2002–2003, the algorithm could predict 75% of the court decisions and 66.7% of the individual votes, whereas the experts correctly predicted 59% of the decisions and 67.9% of the individual votes. Katz, Bommarito and Blackman [42] designed a model based on the Random Forest algorithm to predict the behavior of the U.S. Supreme Court using a time-evolving Random Forest classifier on a corpus of 200 documents, with an accuracy of 70.2%.

For the Turkish language, Faith Sert, Yildirim and Halak [43] use different algorithms to predict court decisions in matters of public morality and freedom of speech using a corpus from the Turkish Constitutional court composed of 92 and 338 legal documents, respectively. They used an embedding representation to perform TF-IDF and Bag-of-Words for inputting the text to a Multi-Layer Perceptron with different architectures. The F-measure results obtained are between 60% and 98.7%. In addition, Virtucio et al. [44] propose using Linear SVM and Random Forest classifiers to predict the decisions of the Supreme Court in the Philippines. They analyzed the Historical Philippine Supreme Court case decision and the Lasphil Project to gather 27,492 cases divided into the following four categories: person, property, public order and drugs. They characterize each use case using the following: case title, case type, year, decision, classification, laws (republic, act, presidential, commonwealth, article, crime) and crime category. They used a subsampling technique to balance negative cases. They used a Bag-of-Words and n-gram representations to model the documents, reaching accuracies of 55% and 59% for Linear SVM and Random Forest, respectively.

Sulea et al. [45] propose predicting the law category, court ruling, and time of the decision of the French Supreme Court. The dataset used for these three prediction tasks was a diachronic collection of rulings from the French supreme court (Court de Cassation) in XML format, containing 126,865 unique documents after the cleaning phase. They use Bag-of-Words, 2-gram and 3-gram as inputs for a linear SVM classifier implemented in Sckit-learn for the different tasks. They report the results using precision, recall, accuracy and F-measure. They sampled 200 documents for the eight different classes in the law prediction task. The F-measure obtained for this task is 90.3%.

For the ruling decision prediction, the SVM algorithm obtained an F-measure of 97% and 92.7% when predicting 6 and 8 classes, respectively. The authors used 1-gram and 2-gram representations for the linear SVM in the last task of temporal prediction, achieving 73.2% and 73.9% when predicting 7 and 14 classes, respectively. Alghazzawi et al. [46] propose a forecasting ruling decision using a hybrid neural network model combining a long-short term memory (LSTM) network with a convolutional neural network (CNN). The authors use 120,506 cases with 27 features from the US Supreme Court ruling. They divide the corpus into 80%, 10%, and 10% for training, validation and testing, respectively. Furthermore, they used 10-fold cross-validation to select the best model. They use oversam-

pling techniques and RFE algorithms to select the best features for handling imbalanced classes and feature selection. After the experiments with different architectures, the best score was an F-measure of 93%.

Based on the Spanish language, Muñoz and Serrano-Soria [47] propose a model to predict court decisions on child custody. First, they study the factors explaining a court's decision to accord child custody by identifying and labeling factual elements and legal principles from the court rulings. Then, they use ANN to predict whether the custody is joint or sole, based on the relationship between the parents and their economic resources, the child's opinion and the psychological report. They used 1884 court sentences from June 2016 to June 2020 from the Spanish Judicial Authority Documentation Center (CENDOJ). The best model output predictions with 85% accuracy.

A limitation of performing court prediction is that access is limited to the annotated corpus. Song et al. [48] gathered a dataset of 50,000 legal opinions and their manually labeled legal procedural postures (POSTURE50K). The dataset were gathered from all states and Districts of Columbia in the United States between 2013 and 2020. The authors propose a DL architecture that adopts domain-specific pretraining and a label-attention mechanism for multi-label document classification based on RoBERTa using this dataset. The authors used POSTURE50K and EUROLEX57K to compare their proposal with other algorithms, such as Random Forest, Multi-Layer Perceptron, BI-GRU-LWAN, AttentionXML, APLC_XLNet and X-Transformer. Depending on the dataset, the F-measure result ranges from 74.5% to 80.2%, depending on the dataset. Thus, in some cases, their proposal outperforms other algorithms. Additionally, Mumcuoglu et al. [49] propose a study to predict decisions in different courts, such as constitutional, civil, criminal, administrative and taxation. They used a corpus consisting of 39,157 legal documents, represented by a reduced Term Frequency vector using the PCA algorithm. The results were used as an input to DT, Random Forest, SVM, Gated Recurrent Unit (GRU), Long Short-Term Memory (LSTMs) and Bidirectional LSTM (BILSTM). Furthermore, they add an attention layer to the GRU, LSTM and BILSTM algorithms. The F-measure obtained results range from 56% to 87%. Sharma, Shandilya and Sharma proposed the eLegPredict system to predict Indian supreme court decisions [50]. The eLegPredict is trained and tested using more than 3072 supreme court cases and the eXtreme Gradient Boosting (XGBoost), ANN, SVM and RF supervised learning algorithms. The system achieved an F-measure between 60% and 76% accuracy (F1-score).

Sharma et al. [51] describe an experiment using deep neural networks and learning techniques to predict decisions of the Supreme Court of the United States with an accuracy of 70.4% on a corpus made up of 7770 cases and 70,000 judicial votes. This experiment is based on research published by Katz, Bommarito and Blackman in 2014, except that they used modern ML techniques instead of using random trees for prediction. RMSprop and Momentum are two techniques to compare results. The proposed approaches include SVM for surface learning and neural networks for DL. The neural network model used is implemented in the Pythons Theano library. They used the dropout technique to obtain results with the DL algorithm that were then compared with surface learning algorithms, such as SVM and complicated neural networks. The authors chose linear, polynomial and nonlinear kernel models for SVMs. The results were encouraging, as prediction levels were even higher than the model proposed by Katz, Bommarito and Blackman. The Katz, Bommarito and Blackman model reached 69.7% accuracy, while Sharma et al. reached 70.2% (Table 2).

Douka et al. [52] propose the JuriBERT pre-trained structure using 123361 documents from the Court of Cassation and Légifrance. The authors use different versions of JuriBERT to classify legal text into eight classes among the chambers and sections and to classify legal text into five categories. The accuracy ranges from 79.9% to 83.28% and 70.38% to 72.09%, respectively.

The work of Sivaranjani, Jayabharathy, and Teja [53] predicts the Indian supreme court decision on appeal cases using a Hierarchical Convolutional Neural Network (HCNN). The

authors used the sentences represented using Word2Vec between 2000 and 2019 to train the Hierarchical CNN, achieving an accuracy ranging from 77.65% to 81.13%.

**Table 2.** Summary of ruling decision prediction in Constitutional Law and Supreme Courts cases, where E = English, Fa = Farsi, F = French, Ph = Philippine, S = Spanish, and T = Turkish.

| Nro. | Reference | Corpus Size | Lang. | Model | Repr. | Metric | Min(%) | Max(%) | Year |
|---|---|---|---|---|---|---|---|---|---|
| 1 | [41] | 628 | E | DT | Binary | Accuracy | 68 | 75 | 2004 |
| 2 | [51] | 120,506 | E | LSTM + CNN | Embeddings | Accuracy | 88 | 92.05 | 2015 |
| 3 | [42] | 28,000 | E | RF | Term frequency | Accuracy | 70.2 | 71.9 | 2017 |
| 4 | [45] | 430 | T | SVM | Bag-of-Words, 2-gram, 3-gram | F-measure | 63 | 90.3 | 2017 |
| 5 | [44] | 27,492 | Ph | SVM, RF | BoW, n-grams | Accuracy | 55 | 59 | 2018 |
| 6 | [49] | 39,157 | T | GRU, LSTM, and BILSTM | TF-IDF | F-measure | 56 | 87 | 2021 |
| 7 | [47] | 1884 | S | ANN | Binary | Accuracy | 61 | 85 | 2021 |
| 8 | [43] | 430 | | MLP | Embedings, TF-IDF, BoW | F-measure | 60 | 98.7 | 2021 |
| 9 | [48] | 50,000 | E | RoBERTa, RF, MLP, BIGRU-LWAN, AttentionXML, APLC_XLNet, and X-Transformer | Sequence of words | F-measure | 74.5 | 80.2 | 2021 |
| 10 | [53] | - | E | HCNN | Accuracy | 77.65 | - | 81.13 | 2021 |
| 11 | [50] | 3072 | E | XGBoost, ANN, SVM, and RF | TF-IDF | F-measure | 60 | 76 | 2021 |
| 12 | [52] | 123,361 | F | JuriBERT | Tokenized | Accuracy | 70.38 | 83.28 | 2021 |
| 13 | [46] | 120,506 | E | CNN, LSTM | Embedding | F-measure | 79 | 93 | 2022 |

### 3.3. Criminal Law

For Criminal Law, reviewed studies used text mining and supervised ML models for managing Judicial files, specifically for treating written criminal sentences in a way that allows the identification of cases. Similar to their classification.

In [54], the authors propose a 7-step methodology to classify the accused person's Acquittal and Conviction (classes) from data obtained from criminal cases related to the murder of the Delhi District Court. Thus, 86 cases were vectorized using the Bag of Word strategy and normalized using the min–max method. Then, algorithms such as SVM, kNN, CART, and NB were used. Finally, leave-one-out cross-validation is used to measure the model's effectiveness, obtaining an accuracy of 85–92% and an F1 score of 86–92%. Ferreira performed another study in Portuguese, and Seron [55]. They compared Logistic Regression (LR), Latent Dirichlet allocation (LDA), KNN, Regression Tree (RT), Gaussian NB (GNB), and SVM algorithm using a corpus consisting of 1562 homicides and corruption in Brazilian court rulings. They obtained an F-measure result ranging from 78% to 98%, depending on the algorithm and court document.

Chou and Hsing [56] developed a document classification, clustering, and search methodology based on neural networks to help law enforcement departments handle written criminal judgments more efficiently. Thus, judges can review similar cases and have a broader view of issuing a sentence. The corpus used to train the models is composed of 210 written criminal sentences from the judiciary of Taiwan for training and testing. Seven crime categories were selected, including homicide, sex crimes, drug-related corruption, computer crime, theft, and fraud. The critical terms used in the trials and files were summarized using 2604 keywords in which 100 main words with the highest frequency of use were selected. For the neural network, 140 sample documents were used. A word segment process was used in the seven specific criminal categories, which was then increased to 251 keywords. Based on the results, the precision obtained reached 94% in the training samples that used all the segments of the written vectors and were not weighted as Back-Propagation Network (BPN) inputs. Similarly, an accuracy of 67% was achieved in the model that uses all the segments of the written vectors, but weighted vectors were used equally as BPN inputs.

Text mining has also been conducted in India. Kaur et al. [57] created a corpus from the police department data downloaded from official pages. Classifiers such as KNN and K-Means clustering algorithm were used. These algorithms were used to evaluate the incidence of crime in the following cities: New Delhi, Andhra Pradesh, Jammu, Kashmir, Daman, Diu, Jharkhand, Arunachal Pradesh, and Nhaveli, of which Delhi and Jharkhand were ranked as the cities with the highest incidence of crime. In the case of Dehli, the result is explained by the city's cosmopolitan nature, the lower availability of preventive measures, and the inability of law enforcement. In the case of Jharkhand, the study concludes that the crime incidence is because the area is too remote for police presence. The authors used Root relative squared error (RRSE) to measure the quality of the KNN algorithm, which obtained 67.92%.

Bingfeng et al. [58] predicted the charges of crimes, such as fraud, theft, or homicide. They used both the legal articles of statutory norms and judicial decisions. To accomplish this, they compiled judicial documents available on the website of the Chinese government, where they extracted descriptions of facts, articles of law, and the charges of imputation. The corpus comprised 50,000 randomly selected documents, 5000 documents for validation, and 5000 documents for testing. The researchers propose a neural network to predict the indictment charges and the relevant articles that serve as the legal basis to justify such charges. The results obtained show the effectiveness of their model for both tasks. A BI-GRU sequence encoder was used in the experiment, an SVM classifier was used for legal article extraction, and SGD was used for training.

Xiao et al. [59] present CAIL2018, a tool that condenses more than 2.6 million criminal cases obtained from the website of the Chinese Supreme People's Court. This tool uses DL techniques to perform NLP tasks combined with neural networks to predict the results of a trial (PJL—Legal Trial Prediction). They used the following three methods in their experiments: TF-IDF + SVM, to extract word features and SVM with a linear kernel to train the classifier. Fast Text was also used to classify texts based on n-grams and Hierarchical softmax. Finally, CNN was used for text classification and fact description coding. Many results were obtained in terms of predicting imputation charges and law articles with high accuracy.

In the same context, Zhong et al. [60] compare text representations, such as TF-IDF and Embeddings with different classification algorithms, such as SVM, CNN, and HLSTM, in over 1.2 million documents of criminal cases in China. The authors used the accuracy to measure the performance of the algorithms, obtaining an accuracy between 38.3% and 94.4%. Furthermore, Li et al. [61] evaluate the judgment rationality of the target case via the judgment results of its similar precedent cases. The authors used Doc2Vec to represent the 41 418 documents in Chinese, which are the input to a GRU. To evaluate the proposal's performance, the authors used the F-measure, which ranges between 73.6% and 78.7%.

Strickson and De La Iglesia propose a work to predict legal decisions or judgments from the UK [62]. The authors used 188 294 English documents to train supervised learning

algorithms, such as SVM, LR, RF, KNN, SLP, and MLP. The authors use different textual representations, such as Count, TF-IDF, LDA, and Embeddings, to compare the algorithms, and they used the accuracy to measure the performance, reaching a value between 49.6% and 69.1%.

In the same spirit, the authors of [63] propose the Law-Article-Distillation-based Attention Network (LADAN) algorithm to face the problem of legal judgment predictions. The proposal includes a graph distillation operator to identify discriminative features from law articles. First, the authors built the similarity using the cosine metric and the TF-IDF vector representation. Later, the authors represent that information in a graph. Then, their proposal LADAN was used, and the results were compared with several baseline algorithms. The authors use three legal datasets and the prediction task's accuracy ranges between 81.20% and 96.60%.

Mahmoudi et al. [64] propose a zork to predict judicial decisions on French courts using an algorithm named CamemBERT judicial. Authors rely on 503 documents of different French courts using FastText, Elmo, BERT, DeLFT, and Word2vec to represent judicial documents. Therefore, they used CamemBERTjudicial and BiLSTM-CRF to predict the courts' decisions. For the experiments, the author reaches results between 81.43% and 100% in terms of the F-measure.

### 3.4. Tax Law, Intellectual Property, and Administrative Law

This section analyses tax, intellectual property, and administrative laws. Text classification in these domains uses corpus resolving tax disputes in countries such as Germany, Brazil and the Netherlands. In intellectual property, the works deal with conflicts over domain names.

For studies on Tax Law, Waltl et al. [65] implemented a Naive Bayes (NB) classifier trained on a corpus of 5990 documents with 11 features, with the objective of predicting whether an appeal was successful in a restricted set of cases from the German taxation court; an F-measure between 53% and 58% was obtained. De Ronde [66] proposes a transformer-based model (RobBERT) to predict Dutch fiscal case-law articles into taxation subareas. A corpus of 59,671 documents was gathered from rechtspraak (rechtspraak.nl: www.rechtspraak.nl, accessed on 29 March 2022) by filtering only those tagged as *Tax Law*. Each document has ten features, 15 subareas of taxation, and a list of contextual keywords for each subarea of taxation. The authors used BERT and RoBERTa and manual labeling for 100 documents concerning the labeling process. For evaluation purpose splitting, the dataset was divided into 80% for training and 20% for validation. Thus, they obtained an accuracy of 87% in the testing set.

For intellectual property, Brangting et al. [67] proposed a methodology to predict and explain decisions in the intellectual property domain. The proposed method is divided into four steps. The first step manually annotates a subset of 16,040 domain name dispute decisions proposed by the World Intellectual Property Organization (WIPO). In the second step, the manually annotated tags are projected using fast text embedding to group semantically similar texts in the annotated and non-annotated corpus to infer the semantics of different tags in the findings section of the decisions. Thus, step 3 builds a mode for each tag using the SCALE algorithm. Finally, step 4 uses an SVM algorithm to predict decisions based on projected features, facts, and contentious texts, achieving an F-measure of 79.5% and 92.2%, respectively. The authors obtained an accuracy of 58.8%.

For Administrative Law, Vihikan et al. [68] developed an automatic resolution of a domain name dispute system. They used an English corpus of 30,311 cases of domain name disputes emitted by the WIPO arbitration and mediation center to implement this system. The author used pretrained models, such as BERT, out-of-the-box BERT (ooBERT), fine-tuned BERT (ftBERT), and LEGAL-BERT. In addition, they used bidirectional GRU and bidirectional LSTM (BILSTM) with the GLOVE pretrained word vectors. Finally, they used logistic regression and linear-kernel SVM model over-weighted TF-IDF feature vectors. The authors obtained an F-measure value between 75.5% and 81.3%. Lage-Freitas, Allende-Cid,

Santana, and de Oliveira-Lage [69] proposed a prototype based on Python's NLTK library to predict legal decisions of Brazilian courts of justice in a corpus of 4762 decisions with a TF-IDF representation, and a result of 78.99% F1-measure was obtained.

Similarly, in [70], the authors propose a complete pipeline for the Connecticut Civil Court Data classification, including Court Administrative Data and Complaint Documents. They focus on several steps in the knowledge extraction from the law textual data process. First, the authors concentrate on the feature engineering step through TF-IDF to obtain lower dimensional features to predict legal cases. Furthermore, embedding such as doc2vec and law2vec were included in the comparison. Then, ML algorithms such as AdaBoost, decision trees, gradient boosting, random forests, SVM, and XGBoost were compared. The metrics used for evaluation were accuracy, precision and recall. Results range between 0.5661 and 0.6274 in terms of the precision–recall ratio.

Chen and Eagle in [71] propose an RF to classify grant or deny of asylum demands. They used 0.5 million documents from 1981 to 2013 with 43 features to reach an accuracy ranging from 75% to 82%.

In [72], the authors built a system called ILDC (Indian Legal Documents Corpus) made up of the annotated judgments (with original explanations) issued by the Indian Supreme Court. The database contains 34,816 documents (from 1947 to 2020), the largest corpus of sentences compiled in the Indian environment. The objective of this research was to develop a system that assists Indian judges in resolving conflicts, suggesting results of similar judicial cases, aiming to speed up judicial processes and reduce delays and procedural overload. The results they obtained from the judicial decision prediction tests were 78% compared to 94% obtained by human legal experts. To deal with court case information, the authors identified some regular expressions to remove noisy text and meta information, for example, the initial parts of the document containing the case number, judge name, dates and other process meta-information. As the authors indicate, this is because the judge who decides the case influences the final decision. The authors also worked with processes with multiple claims, wherein, in a single case, the appellant presented multiple requests, which led the authors to divide the cases into two groups. One set is made up of cases with a single decision, and another is made up of appeals with different decisions based on multiple claims.

In [73], the authors experimented with different types of deep neural networks applied to the classification of legal consultations. As the authors point out, legal query comprehension is a complex problem that involves two Natural Language Processing (NLP) tasks that must be solved together: (i) identifying user intent and (ii) recognizing entities within queries. Thus, the authors used deep neural architectures as well as Recurrent Neural Networks (RNNs), Long Short Term Memory (LSTM), Convolutional Neural Networks (CNNs) and Gated Recurrent Units (GRU) and compared them both individually and in combinations. The models were also compared to Machine Learning (ML) and rule-based approaches. The results of these experiments show that it is difficult for DNNs to deal with long queries.

### 3.5. Private Law

To the best of our knowledge, few documents have been found that report the use of machine learning and deep learning applied to private law. Indeed, two main contributions were found in this context.

Li et al. [74] propose a Markov Logic Networks (MLN) probability model to predict the judicial decision of divorce cases. The authors used 695 418 documents from China Judgments Online (China Judgments Online: wenshu.court.gov.cn/, accessed on 16 September 2022). From these documents, authors extract confirmation of facts, articles and judicial decisions to train the MLN. They could predict the probability of a granted divorce (89.7%) and the plaintiff paying the fees (87.06%), with an F1-measure between 73.58% and 77.74%.

Other papers face the problem of class limits between the prediction of one class instead another class in the classification problem. Thus, in [75], the authors propose a

two-layered hierarchical fuzzy algorithm applied to a dataset containing Iranian standard contracts between an employer and a contractor grouped into three classes: building, semi-building and non-building. Before applying the algorithm, the authors performed a feature selection step using the normalized entropy measure. Later, the corpus was used as input to the algorithm to extract fuzzy rules, and the accuracy was measured using the 10-fold cross-validation technique.

In [76], the authors processed a collection of court decisions in French from the Régie du Logement du Québec (RDL) on real estate law litigation. Indeed, on a corpus made up of 981,112 decisions issued from 2001 to 2018 by 72 judges in 29 Quebec courts, the authors applied NLP tools to reveal the biases that can influence prediction experiments. The authors divided the rulings into two broad categories: Landlord versus Tenant (LvT) and Tenant versus Landlord (TvL). In this experiment, they used the FlauBERT language model. They managed to identify more than a dozen characteristics of each decision and detect biases contained in the data sets, such as, for example, that landlords tend to sue their tenants, with more probability of success. The prediction results are 93.7% and 85.2% in LvT cases and 84.9% and 74.6% for TvL cases, respectively.

## 4. Quantitative Analysis of Judicial Decision Prediction Studies

This section presents the analysis performed on the gathered articles. The first analysis performed was about the corpus size over the years, as shown in Figure 2. The maximal size of the corpus is shown in blue, and the average corpus size is shown in green. Note that the corpus size increased over the years, reaching 100,000 documents analyzed in 2020. It is important to note that the corpus with the largest size (>5.5 million) was used in 2018.

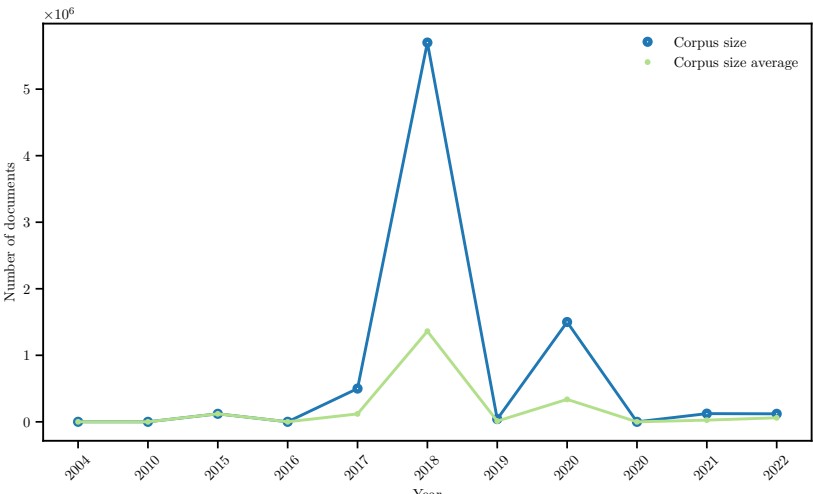

**Figure 2.** Corpus size evolution.

Regarding the documents, the languages used for the judicial decision prediction are English, German, Portuguese, Turkish, Chinese, Farsi, French, Iranian and Spanish. English is most used language in the literature (c.f., 60%), as depicted in Figure 3. Other languages are Chinese and Portuguese, representing 10% and 6.7% of the works, respectively. Finally, the other languages represent 23.3% of the works in the literature review.

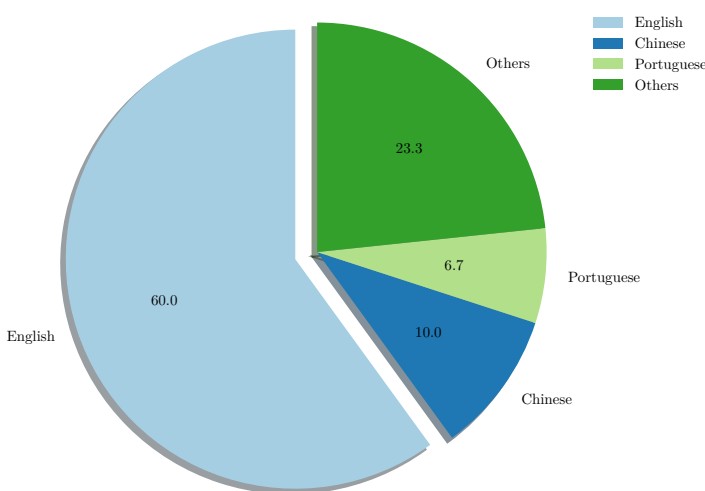

**Figure 3.** Languages distribution of analyzed works.

For the learning model, the different techniques used for predicting rule decisions are shown in Figure 4a. Note that data mining techniques, such as SVM, K-NN and RF are the most used. Furthermore, the DL techniques frequently used are CNN, LSTM and transformers, such as BERT.

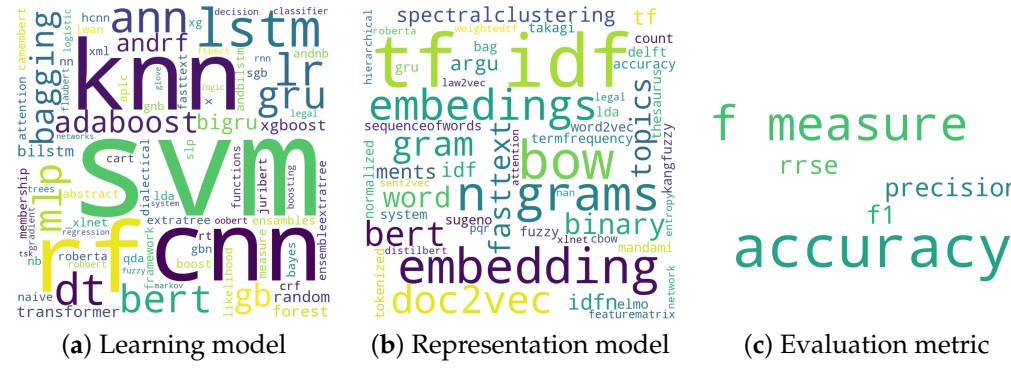

| (**a**) Learning model | (**b**) Representation model | (**c**) Evaluation metric |

**Figure 4.** Word cloud analysis.

Figure 4b illustrates the models used for the textual representation model. Term Frequency, Inverse Document Frequency (TF-IDF), n-grams, Bag-of-Words (BoW) and embedding are the most used. Apropos evaluation metrics, Figure 4c shows that Accuracy and F-measure are the most used metrics to evaluate the learning model's performance in predicting the judicial decision.

For the performance analysis, the increasing trend in the performance of the judicial decision prediction, even when the corpus size also increases, is shown in Figure 5. This trend is due to the shift from ML to DL techniques, which handle a significant corpus achieving impressive performances. Nonetheless, the predominance of ML over DL techniques is shown in Figure 6. Note that DL techniques have been more popular since 2021. Finally, the minimal and maximal values for different metrics used to assess the performance of ruling decision predictions are illustrated in Figure 7.

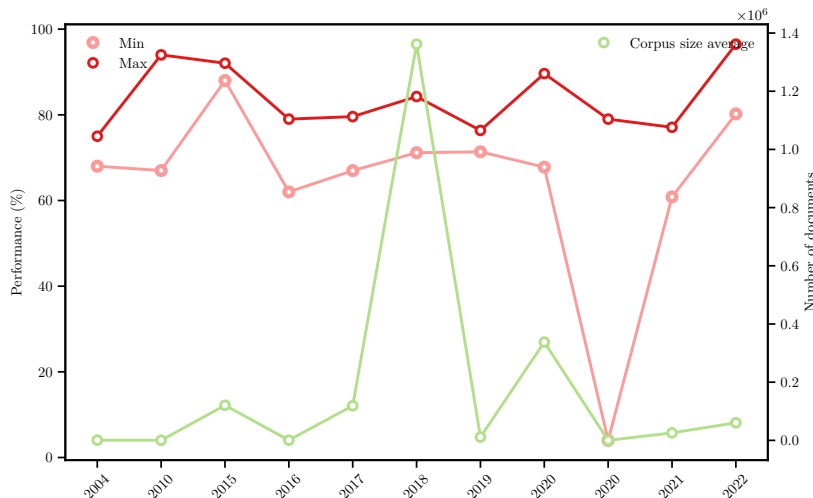

**Figure 5.** Historical performance of judicial decision prediction.

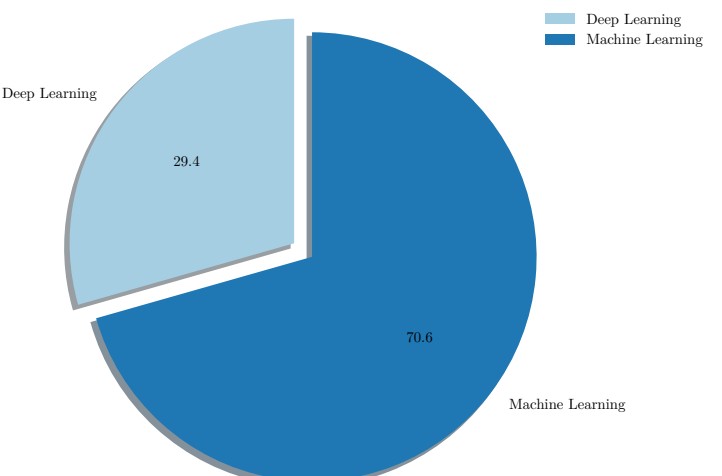

**Figure 6.** Percentage of models used for ruling decision prediction.

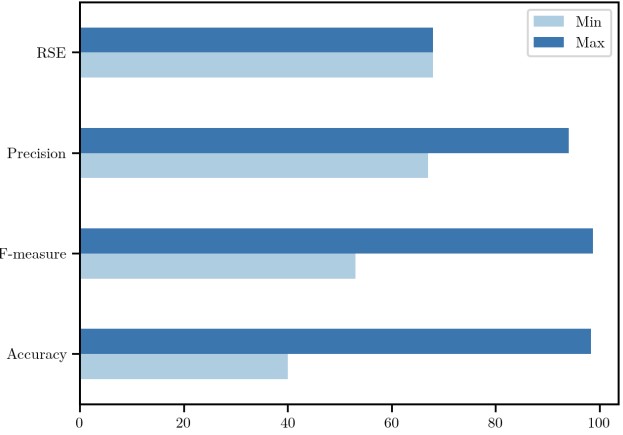

**Figure 7.** Scores using different metrics to assess the ruling decision prediction.

## 5. Discussion

This section discusses our findings in the state-of-the-art following the question proposed in Table 1. To achieve this goal, information from articles gathered in this study (c.f., Section 3) and the quantitative analysis of tables summarizing their main characteristics (c.f., Section 4) were used.

### 5.1. Which Law Type Applies Judicial Decision Prediction through ML or DL?

The state-of-the-art shows that many NLP methods have been used for analyzing textual corpus associated with the legal domain. After applying the literature search methodology, eight hot topics were revealed, such as Human Rights, Constitutional Law, ruling prediction of Supreme Courts, Tax Law, Intellectual Property, Administrative Law, and Criminal Law. For our purpose, we grouped these topics into four categories in Section 3. For the performance analysis, the increasing trend in the performance of the judicial decision prediction, even when the corpus size also increases, is shown in Figure 5. This trend is caused by the shift from ML to DL techniques, which handle significant corpora, achieving impressive performances. Nonetheless, the predominance of ML over DL techniques is shown in Figure 6. Note that DL techniques have been used more since 2021. Finally, the minimal and maximal values for different used metrics for evaluating the performance of ruling decision predictions are shown in Figure 7.

### 5.2. Which NLP and ML Techniques Were Used to Analyze Textual Data Associated with the Legal Text?

Tables 2–6 summarize the NLP techniques used for analyzing textual data in the legal domain.

**Table 3.** Summary of ruling decision prediction in Human Rights, where E = English.

| Nro. | Reference | Corpus Size | Lang. | Model | Repr. | Metric | Min(%) | Max(%) | Year |
|------|-----------|-------------|-------|-------|-------|--------|--------|--------|------|
| 1 | [33] | 584 | E | SVM | BoW, n-grams, topics | Accuracy | 62 | 79 | 2016 |
| 2 | [36] | 584 | E | KNN, LR, Bagging, RF, SVM | Topics | Accuracy | 66.5 | 87.5 | 2017 |
| 3 | [38] | 1942 | E | SVM | TF-IDF n-grams | Accuracy | 66 | 77 | 2018 |
| 4 | [37] | 9703 | E | GB, RF, SGB, DT, and QDA | n-grams. word embeddings and doc2vec | Accuracy | 68.83 | 68.83 | 2019 |
| 5 | [39] | 684 | E | SVM, GBN, KNN, Ensambles | BoW + Spectral clustering | Accuracy | 69.6 | 86.1 | 2019 |
| 6 | [40] | 18,420 | E | AdaBoost, Bagging, NB, DT, Ensemble Extra Tree, Extra Tree, GB, KNN, SVM, NN, and RF | TF-IDF, n-grams | Accuracy | 75.86 | 98.32 | 2020 |
| 7 | [35] | 10 | E | Abstract Dialectical Framework | Arguments | Accuracy | - | 79 | 2020 |
| 8 | [34] | 11,532 | E | SVM | TF-IDF | Accuracy | 77 | 79 | 2020 |

First, several efforts were proposed for Human Rights. All studies reviewed present a binary classifier. Some attempt to classify violation or non-violation of an article from the ECHR. The corpus was vectorized using strategies, such as TF-IDF, n-grams, Bag-of-Words, word embedding, and paragraph embeddings. SVM was the most popular used supervised learning algorithm. Other algorithms were KNN, NB, CART, and various Random Forest variants.

For analyzing documents concerning constitutional law and ruling prediction of Supreme Courts, most studies address the problem of binary classification using classical ML algorithms, such as decision trees, RF, SVM, and XGBoost. Furthermore, recent DL techniques were used, such as CNN, LSTM, and their variants, such as BiLSTM, BI-GRU, Attention, and RoBERT. The authors used classical techniques for text vectorization, such as

binary, TF, TF-IDF representation, embeddings, and fuzzy representation. Finally, accuracy and F-measure measures were used to measure the effectiveness of the classification task.

Regarding Tax Law, Intellectual Property, and Administrative Law, algorithms such as NB, SVM, XGBboost, decision trees, logistic regression, BERT, Transformers, CNN, and LSTM, were used for the prediction task. Furthermore, some authors proposed modifications to the BERT algorithm, such as ooBERT, LEGAL-BERT, and ft-BERT. The TF-IDF vector representation of documents is used. In addition to these techniques, two adapted ones were used to improve results, such as normalized thesaurus and weighted TF-IDF. The F-measure is the most used evaluation measure in this category.

**Table 4.** Summary of ruling decision prediction in Tax Law, Intellectual Property, and Administrative Law, where E = English, F = French, G = German, P = Portuguese and I = Iranian.

| Nro. | Reference | Corpus Size | Lang. | Model | Repr. | Metric | Min(%) | Max(%) | Court |
|------|-----------|-------------|-------|-------|-------|--------|--------|--------|-------|
| 1 | [65] | 5990 | G | Naive Bayes classifier | norma- lized thesaurus | F-measure | 53 | 58 | 2017 |
| 2 | [71] | 500,000 | E | RF | feature matrix | Accuracy | 75 | 82 | 2017 |
| 3 | [73] | 545,760 | E | CNN, LSTM, RNN | CBoW | F-measure | 99.74 | 99.89 | 2018 |
| 4 | [69] | 4762 | P | SVM, DT, XGBoost, BERT, GRU, LSTM, CNN | TF-IDF | F-measure | 73.42 | 80.22 | 2019 |
| 5 | [66] | 53,397 | G | Transformers, BERT RobBERT | PQR | Accuracy | 61 | 87 | 2021 |
| 6 | [67] | 16,040 | E | SVM | Embedings | F-measure | 79.5 | 92.2 | 2021 |
| 7 | [68] | 30,311 | E | ooBERT, ftBERT, and LEGAL-BERT, biLSTM, GRU, LR, SVM | weighted TF-IDF | F-measure | 75.5 | 81.3 | 2021 |
| 8 | [70] | 7904 | E | adaBoost, decision trees, gradient boosting, random forests, SVM, XGBoost | TF-IDF, doc2vec, law2vec | Accuracy | 56.61 | 62.74 | 2021 |
| 9 | [72] | 34,816 | I | Logistic Regression, SVM, Random Forest, BiGRU, GloVe, | Sent2Vec, Doc2Vec, Hierarchical Attention Network, BERT, DistilBERT, RoBERTa, XLNet, LEGAL-BERT | Accuracy | 78 | 94 | 2021 |

*5.3. What Are the Most Used Techniques in ML and DL to Predict Judicial Decisions?*

Even though ML and DL fall under the AI umbrella, some differences exist. Supervised ML extracts patterns from vectorized textual data to classify and improve the experience. DL is a subset of ML that uses complex, multi-layer neural networks. Even though ML and DL fall under the AI umbrella, DL networks have a significant benefit in that they frequently improve as the quantity of data grows [77].

For the classification of legal documents, several traditional ML techniques, such as NB, Logistic Regression, SVM, RF, and variations MLP, and XGBoost, were used. Most of them were conducted well in terms of the classification ratio or accuracy. The DL methods have also been used for legal textual data classification. Algorithms such as LTSM, CNN, BERT, and other variations were successfully used in the literature. The percentage of ML (70.6%) and DL (29.4%) techniques used in the works in the literature is shown in Figure 6.

The best result was obtained using the ML algorithm (Multi-Layer perceptron) with 98.7%.

### 5.4. Which Countries Apply or Implement Judicial Decisions Prediction through ML or DL?

As shown in Figure 3, most studies were written in English (more than 60%). Furthermore, Portuguese has a representative number of publications addressing the problem of legal texts classification. Nevertheless, Turkish, Chinese, Spanish, Philippine, and Farsi are other languages.

### 5.5. Which Quality Measures Are Used to Validate Decision Prediction?

AI is invaluable in automating time-consuming tasks. Nevertheless, the credibility of the models is always debatable [77]. To address this issue, it is crucial to measure the performance of algorithms in terms of the success ratio. In addition, it is important to have as few errors as possible in the classification (false positives and true negatives). The performance of an algorithm depends on multiple factors, such as the corpora used, the application domain, and the ML algorithm. The literature reviewed in our study reveals the use of several quality measures. The most commonly used is the accuracy, which measures the success classification ratio. Other measures include the F1 score, precision, and RSE. The percentage of quality measures found in the literature is shown in Figure 7.

### 5.6. Which Law Type Uses the Most Judicial Decisions Prediction through ML or DL?

As shown in Tables 2–5, Constitutional Law and Ruling Prediction of Supreme Courts is the category with the highest number of applications using ML or DL. The second one is related to Criminal Law, and only a few studies were found for the category combining Tax Law, Intellectual Property, and Administrative Law.

**Table 5.** Summary of ruling decision prediction in Criminal Law, where C = Chinese, E = English, F = French, P = Portuguese and S = Spanish.

| Nro. | Reference | Corpus Size | Lang. | Model | Repr. | Metric | Min(%) | Max(%) | Year |
|---|---|---|---|---|---|---|---|---|---|
| 1 | [56] | 210 | C | ANN | TF-IDF | Precision | 67 | 94 | 2010 |
| 2 | [58] | 60,000 | C | SVM, ANN | Embedding | F-measure | 70.15 | 98.51 | 2017 |
| 3 | [59] | 5.7 M | C | SVM, Fast Text, CNN, | TF-IDF, n-grams, embeddings | Accuracy | 94.3 | 97.6 | 2018 |
| 4 | [60] | 1.2 M | E | SVM, CNN, HLSTM | TFIDF, Embedings | Accuracy | 38.3 | 94.4 | 2018 |
| 5 | [61] | 41,418 | C | GRU, likelihood measure | Doc2Vec | F1 | 73.6 | 78.7 | 2019 |
| 6 | [57] | 38 | E | KNN | BoW | RRSE | | 67.92 | 2019 |
| 7 | [62] | 188,294 | E | SVM, LR, RF, KNN, SLP, MLP | Count, TF-IDF, LDA, Embeddings | Accuracy | 49.6 | 69.1 | 2020 |
| 8 | [54] | 86 | E | SVM, kNN, CART, and NB | BoW | F-measure | 86 | 92 | 2020 |

**Table 5.** *Cont.*

| Nro. | Reference | Corpus Size | Lang. | Model | Repr. | Metric | Min(%) | Max(%) | Year |
|------|-----------|-------------|-------|-------|-------|--------|--------|--------|------|
| 9 | [63] | 1.5 M | C | LADAN | TF-IDF | Accuracy | 81.20 | 96.6 | 2020 |
| 10 | [55] | 1562 | P | LR, LDA, KNN, RT, GNB, SVM | TF-IDF | F-measure | 78 | 98 | 2020 |
| 11 | [64] | 503 | F | CamemBERT, BiLSTM-CRF | FastText, Elmo, BERT, DeLFT, Word2vec | F-measure | 81.43 | 100 | 2022 |

**Table 6.** Summary of ruling decision prediction in Public Law, where C = Chinese, F = French and I = Iranian.

| Nro. | Reference | Corpus Size | Lang. | Model | Repr. | Metric | Min(%) | Max(%) | Court |
|------|-----------|-------------|-------|-------|-------|--------|--------|--------|-------|
| 1 | [74] | 695 418 | C | Markov logic networks | | F-measure | 73.58 | 77.74 | 2018 |
| 2 | [76] | 981112 | F | FlauBERT | TF-IDF, FastText | Accuracy | 74.6 | 93.7 | 2020 |
| 3 | [75] | 100 | I | TSK fuzzy system | Entropy | Accuracy | 55.3 | 58.8 | 2021 |

## 6. Conclusions

In recent years, artificial intelligence, specifically text mining, has been applied in many areas of knowledge and in systems that humans use daily (chatbots, voice recognition, etc.). The application of these technologies in the legal domain, especially in predicting judicial decisions, has been increasing in recent years. The reviewed papers show, in this ambit, a predominance of the use of text mining techniques, such as Support Vector Machine (SVM), K Nearest Neighbors (K-NN) and Random Forest (RF), over deep learning techniques. These techniques have been applied to predict judicial decisions in various legal domains, such as human rights, divorces, intellectual property, etc., and used to predict the commission of crimes, impute charges or identify the areas of cities with the highest incidence of crime. Although NLP methods are used in the analysis of the legal textual corpus, this trend is due to the shift from ML to DL techniques, which handle a significant corpus and achieve impressive performance. However, the predominance of ML techniques over DL is observed in the results obtained in the works reviewed, i.e., the percentage of ML was 70.6%) and DL was 29.4%. The best result was obtained using the ML algorithm (Multi-Layer perceptron) with 98.7%. This behavior is observed in constitutional law and sentence prediction since the authors preferred using ML techniques such as decision trees, RF, SVM and XGBoost. Furthermore, recent DL techniques were used, such as CNN, LSTM, and their variants, such as BiLSTM, BI-GRU, Attention, and RoBERT. In other areas of law, such as Tax Law, Intellectual Property, and Administrative Law, algorithms such as NB, SVM, XGBoost, decision trees, logistic regression, BERT, Transformers, CNN, and LSTM, were used for the prediction task. Furthermore, some authors proposed modifications to the BERT algorithm, such as ooBERT, LEGAL-BERT, and ft-BERT. Most of the research has been conducted in English. We found that 64% of the works belong to studies carried out in English-speaking countries, 8% in Portuguese and 28% in other languages (such as German, Chinese, Turkish, Spanish, etc.). Very few works of this type have been carried out in Spanish-speaking countries. Applying these prediction techniques in the different legal domains reaches an accuracy of 60%.

We believe that it would be interesting to extend this work to other thematic areas and focus on the new technologies that are being developed in legal intelligence for future work.

**Author Contributions:** Conceptualization, O.A.A.F., M.N.-d.-P. and H.A.-S.; methodology, O.A.A.F., M.N.-d.-P. and H.A.-S.; validation, M.N.-d.-P., O.A.A.F. and H.A.-S.; formal analysis, O.A.A.F., M.N.-d.-P. and H.A.-S.; investigation, O.A.A.F., M.N.-d.-P. and H.A.-S.; resources, O.A.A.F.; data curation, M.N.-d.-P.; writing—original draft preparation, O.A.A.F., M.N.-d.-P. and H.A.-S.; writing—review and editing, M.N.-d.-P. and H.A.-S.; visualization, M.N.-d.-P. and H.A.-S.; supervision, O.A.A.F., M.N.-d.-P. and H.A.-S.; project administration, O.A.A.F.; funding acquisition, O.A.A.F. All authors have read and agreed to the published version of the manuscript.

**Funding:** This research was funded by the Instituto de Investigación Científica (IDIC), Universidad de Lima.

**Institutional Review Board Statement:** Not applicable.

**Informed Consent Statement:** Not applicable.

**Data Availability Statement:** The code and files for results replication are available at https://github.com/bitmapup/SurveyJudicialDecisionsPrediction.

**Acknowledgments:** This article was developed as part of the research project "Application of Big Data in the "personalization" of consumer law" of the Scientific Research Institute (IDIC) of the University of Lima.

**Conflicts of Interest:** The authors declare no conflict of interest.

## Abbreviations

The following abbreviations are used in this manuscript:

| Abbreviation | Name |
| --- | --- |
| APLC XLNET | Pretrained Generalized Autoregressive Model with Adaptive Probabilistic Label Clusters |
| ANN | Artificial neural network |
| BI-GRU | Bidirectional Gated recurrent units |
| BI-LSTM | Bi-directional Long Short Term Memory |
| BERT | Bidirectional Encoder Representations from Transformers |
| BoW | Back of Word |
| BPN | Back-Propagation Network |
| CART | Classification And Regression Trees |
| CNN | Convolutional Neural Network |
| DT | Decision Tree |
| GB | Gradient Boosting |
| GRU | Gated Recurrent Unit |
| KNN | K-Nearest Neighbors |
| LDA | Linear Discriminant Analysis |
| LSTM | Long Short-Term Memory |
| LR | Logistic Regression |
| LWAN | Label-Wise Attention Network |
| NB | Naive Bayes |
| NN | Neural network |
| ftBERT | Fine-Tuned Bert |
| ooBERT | out-of-the-box BERT |
| MPL | Multi-Layer Perceptron |
| RoBERTa | Robustly Optimized BERT Pretraining |
| RBF | Radial Basis Function |
| SGB | Stochastic Gradient Boosting |
| SLP | Stochastic Logic Programs |
| SVM | Support Vector Machine |
| TF-IDF | Term Frequency Inverse Document Frequency |
| RF | Random Forest |

| QDA | Quadratic classifier |
| XG Boost | eXtreme Gradient Boosting |
| X Transformer | eXtreme Transformer |
| MLN | Markov Logic Networks |
| HCNN | Hierarchical Convolutional Neural Network |
| SRL | Systematic Literature Review |

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
