# Peer review of "Survey of Text Mining Techniques Applied to Judicial Decisions Prediction"

_applsci, doi:10.3390/app122010200_

Round 1

Reviewer 1 Report

The manuscript presents a survey of text mining techniques in the legal domain. The authors report that 64% of the works came from English-speaking countries, 8% from Portugal, and 28% from other countries, and the accuracy of the techniques for judicial decisions prediction is above 60%.

The manuscript discusses an interesting topic but needs a lot of improvement:

1.      The title needs to be reviewed. It is obliged to choose either “Artificial Intelligence Techniques” or "Text mining Techniques" as the Conclusions section mentions?

2.      Classification of survey is missing (with diagram).

3.      Relationship with the previous surveys in this area, is not established.

4.      Research questions were presented, but the results section did not answer these questions in detail.

5.      Keywords and search strings also need further consideration, currently missing a lot of keywords directly related to the topic of the survey. For example:

·         deep learning + judicial decisions prediction

·         NLP + human rights

6.      During 22 years (since 2000), only 30 documents were surveyed, which is not sufficient. This number can make survey results no longer reliable enough.

7.      The comparative tables are very short, lacking the reviewed articles. Table 2 needs to be improved, currently only English language is mentioned.

8.      Table 2: 8. What criteria are for ordering items: publication year, the accuracy…? Same question with Table 3, Table 5.

9.      The open issues and challenges section is not convincing.

10.  The Abstract and The Conclusions should be rewritten for the better. The findings mentioned in the Abstract are poor.

Author Response

Dear Reviewer,

We thank you for your comments. We have carefully considered each of comments and we have accordingly introduced changes in the paper, detailed in the attached PDF file.

--

Hugo Alatrista Salas

The corresponding author

Reviewer 2 Report

The authors discuss a very interesting topic related to the use of artificial intelligence to predict court decisions. Although from my point of view as a practice in this field, it seems that despite the undoubted advantages of AI for this type of analysis processes - de facto appropriate statistical analysis performed by AI, it is difficult to take into account all "ideas" and the views of judges. However, I believe that such a tool would be helpful for the parties to the proceedings to determine or modify the procedural strategy. In turn, ethical aspels are associated with this, because in fact it is influencing a process that, by definition, should be an emanation of justice. Technically, however, this is possible.

Interestingly presented in the Introduction article with good literature appeals.

In addition to the natural components of the article like State of Art, the analysis of prediction deserves to be noticed. The same way and presentation but also the conclusions shown.

Undoubtedly, the discussion is a discussion, which is a discussion and not just an empty chapter with this name. Suggests developing conclusions.

Author Response

(The authors gave the same response as above.)

Round 2

Reviewer 1 Report

Thank you for updating the manuscript according to the comments.